# The Role of Transcription Factors in the Regulation of Plant Shoot Branching

**DOI:** 10.3390/plants11151997

**Published:** 2022-07-31

**Authors:** Lingling Zhang, Weimin Fang, Fadi Chen, Aiping Song

**Affiliations:** State Key Laboratory of Crop Genetics and Germplasm Enhancement, Key Laboratory of Landscaping, Ministry of Agriculture and Rural Affairs, College of Horticulture, Nanjing Agricultural University, Nanjing 210095, China; 2021204041@stu.njau.edu.cn (L.Z.); fangwm@njau.edu.cn (W.F.)

**Keywords:** transcription factors, branching, axillary meristem, development

## Abstract

Transcription factors, also known as trans-acting factors, balance development and stress responses in plants. Branching plays an important role in plant morphogenesis and is closely related to plant biomass and crop yield. The apical meristem produced during plant embryonic development repeatedly produces the body of the plant, and the final aerial structure is regulated by the branching mode generated by axillary meristem (AM) activities. These branching patterns are regulated by two processes: AM formation and axillary bud growth. In recent years, transcription factors involved in regulating these processes have been identified. In addition, these transcription factors play an important role in various plant hormone pathways and photoresponses regulating plant branching. In this review, we start from the formation and growth of axillary meristems, including the regulation of hormones, light and other internal and external factors, and focus on the transcription factors involved in regulating plant branching and development to provide candidate genes for improving crop architecture through gene editing or directed breeding.

## 1. Introduction

Transcription factors (TFs), also known as trans-acting factors, are proteins with special structures that regulate plant growth and development. Transcription factors bind to specific DNA sequences (cis-acting elements) in the upstream promoter region of target genes through their DNA-binding domain (DBD), thereby regulating the specific expression of target genes in different cell types of plants or under different environmental conditions [1]. In plant morphogenesis, selective expression of genes leads to the differentiation of phenotypes, and transcription factors play an important regulatory role in these processes. TFs are divided into different gene families, such as the bHLH, TCP, MADS, bZIP, KNOX, WOX, AP2/ERF, NAC, GATA and ARF families, according to differences in DNA-binding domains and conserved motifs.

Shoot branching is a common phenomenon in plant growth and plays a very important role in plant morphogenesis. Branching also affects plant competitiveness against weeds or pests [2,3]. Therefore, research on branching mechanisms has become a popular topic worldwide. Studies have shown that axillary meristems initiate from cell groups detached from the primary SAM that retain their meristematic identity (Figure 1). Alternatively, axillary meristems may originate de novo later in development from partially or fully differentiated cells. Development of the lateral branch involves two important processes, axillary meristem formation and axillary bud growth [4,5].

Through a study on the regulatory mechanism of plant branching, a series of branching-related transcription factors (Table 1) have been isolated from *Oryza sativa*, *Arabidopsis thaliana*, *Lycopersicon esculentum*, *Zea mays* and other plants. However, the nomenclature of homologous genes in different species is confusing, and related studies on transcription factors involved in the regulation of branching development are lacking in systematic elaboration. Branch formation is regulated at two developmental stages: axillary bud meristem formation and axillary bud emergence. The latter is induced by bud dormancy release and regulated by the synergistic effect of plant hormones such as auxin (IAA), strigolactones (SLs), cytokinins (CKs), abscisic acid (ABA) and brassinosteroids (BRs) [6,7]. Of course, many endogenous and developmental signals can be integrated to determine the fate of buds and the number and location of new buds growing on plants. This regulation is also strongly dependent on environmental factors, and plants adjust their branching ability according to their environmental conditions [8].

In this review, we begin with the formation and growth of axillary meristems to elaborate on the research progress of transcription factors involved in regulating plant branching development to provide target genes for manipulating plant branching.

## 2. Axillary Meristem Formation

The first step of branching is the development of axillary meristems in leaves. In recent years, a series of transcription factors that regulate the initiation of leaf axil meristems have been found in *Arabidopsis*, rice, maize and tomato. The AM is formed in the center of the frontal boundary zone of the leaf base. This region is not only a boundary but also plays an important role in the maintenance of meristem and organ development [9].

The origin of AMs is a controversial topic. There is a major view that AMs originate from meristem cell groups that become detached from the shoot apical meristem (SAM) as leaves form and never lose their meristem identity [10]. Early AM development depends on the maintenance of the specificity and meristem ability of axillary cells. In *Arabidopsis* (Figure 2), *WUSCHEL* (*WUS*) is a homologous domain transcription factor that is expressed in the center of the SAM and specifies the fate of meristem cells in this region. The *wus* mutation leads to an inability to maintain stem cell division ability [11]. ARABIDOPSIS RESPONSE REGULATOR 1 (ARR1) is a transcription factor downstream of CK that promotes *LAS* expression by binding to its promoter, promoting AM initiation [12,13]. Cytokinins also activate *WUS* expression through ARR1, enabling stem cell differentiation and axillary bud formation [14].

*MERISTEMLESS* (*STM*) is another important factor for maintaining branch organization. *STM*, a KNOX gene, is expressed in the whole SAM but is excluded from the organ primordium that maintains the function of undifferentiated cells in the SAM [15]. The molecular markers of the AM include concentrated and strong expression of *STM* in the center of the boundary region [16]. Once cells begin to differentiate, *STM* is downregulated by the MYB family transcription factor ASYMMETRIC LEAVES1 (AS1) and the LBD family transcription factor AS2 [17]. This indicates that the cells in the border region maintain the ability to recover to the meristem stage within a limited period of time. During the developmental stage, AMs began to form [18].

Before axillary bud formation, REVOLUTA (REV) upregulates *STM* expression and promotes AM initiation. Subsequently, CK reactivates *WUS* expression to establish the AM [1]. Preliminary evidence shows that REV acts upstream of *STM* and *WUS* and that Ls/LAS acts upstream of *STM* to activate expression [19,20]. However, the upstream region of *REV* is regulated by LAS. As an HD-ZIP transcription factor, REV itself is necessary for all lateral meristem formation. In addition to RNA accumulation in other modes, *REV* is expressed in the near-axis position of the developmental leaf primordium in a region similar to *RAX1*, which can produce the position signal of *RAX1* expression and control the radial mode [21,22]. The AP2 family transcription factor DORNRÖSCHEN (DRN) also plays a role in embryonic meristem and lateral organ development. Although AM initiation is affected in the *drn-1* mutant, the *drn-1 drnl-1* double mutant shows more serious defects in axillary bud formation than the single mutant, indicating that *drn* and *drnl* have important redundant functions in AM initiation [1]. Further studies have shown that DRN and DRNL preferentially affect the initiation pathway of the AM at the early nutritional stage rather than at late and reproductive stages. DRN, DRNL and REV can directly activate *STM* expression by binding to the same promoter region. In summary, DRN and DRNL redundantly promote AM initiation at the vegetative growth stage, and DRN/DRNL and REV synergistically upregulate *STM* transcription in mature leaf axils [23].

REGULATOR OF AXILLARY MERISTEM FORMATION (ROX) encodes direct homologs of bHLH transcription factors, namely, LAX1 and BA1. In these mutants, axillary bud formation during vegetative bud development is damaged, and their combination with REGULATOR OF AXILLARY MERISTEMS1 (*rax1*) and *las* mutations enhance these branching defects [24], indicating that ROX regulates AM formation by cooperating with RAX1 and LAS [18]. In the nutritional and reproductive development process, orthologous bHLH transcription factors seem to be involved in the formation of boundary regions in eudicotyledons and Gramineae plants, as delayed growth is needed. In subsequent studies, ABA was found to significantly inhibit the expression of the *RAX1* and *LAS* genes, thereby affecting the growth of axillary buds. RAX1 is a member of the largest MYB TF R2R3-MYB family in *Arabidopsis*. *RAX2* and *RAX3* genes function in the early stages of AM initiation and development [22,24,25]. In addition, RAX1 promotes the early stage of AM formation; it also negatively regulates the gibberellin level in shoot tips to modulate AM formation and affect the timing of development phase transition [26]. Studies have shown that RAX1 is involved in the determination of axillary meristems by generating a tissue environment conducive to the establishment of meristems to control the spatial pattern of AM development [22]. In sunflower, the rax-like gene R2R3-MYB2 was also found to play a key role in AM formation to establish or maintain the leaf axillary stem cell niche [24]. The three genes are expressed at the regional boundary between the shoot apical meristem and leaf primordium prior to the establishment of axillary meristem [22]. Further evidence shows that LAS and RAX1 can replace ROX to some extent and regulate axillary meristem formation [27].

LATERAL ORGAN FUSION1 (LOF1) and LOF2 encode MYB transcription factors that play roles in lateral organ separation and axillary meristem formation, partly through interaction with CUC2, CUC3 and STM [16,28]. *LOF1* is expressed at organ boundaries and acts upstream of *RAX1*, *LAS* and *CUC* [16,29,30]. The *lof1* mutant exhibits defects in organ separation, which is the result of abnormal cell division and amplification during early boundary formation. In addition, low concentrations of BRs in the border area promote specific expression of the *CUC* gene and initiation of the AM [21].

The NAC domain proteins CUP SHAPED COTYLEDON1 (CUC1), CUC2 and CUC3 are involved in the initiation of the *Arabidopsis* SAM via *STM* expression [16,22,27,28]. The number of axillary buds in its mutants is significantly decreased [31]. Among the three CUC family members, CUC3 plays a major role in determining the formation and location of axillary meristems [30,31,32]. Further studies have shown that CUC1 and CUC2 control the development of the axillary meristem by regulating LAS and that CUC3 may play a role independent of LAS [30]. *CUC2* is also regulated by RAX1 in early AM development to establish or maintain the stem cell niche formed by the AM [22,31]. In particular, transcription of *CUC2* is continuously downregulated in the *raxl* mutant, indicating that RAX1 affects AM initiation by regulating the expression of *CUC2* [21]. Thus, RICE FLORICULA/LEAFY (RFL) promote AM specificity through an effect on *LAX1* and *CUC* [33]. *RFL* is expressed in the vegetative axillary meristem and very young tillering buds, and its expression pattern is similar to that of STM, which may be related to the maintenance of the meristem cell zone [34].

Guo et al. [35] identified the *EXCESSIVE BRANCHES1* (*EXB1*) gene, which encodes a WRKY transcription factor previously known as WRKY71 that is mainly expressed in tissues around the AM initiation site. The functional *exb1-D* mutant displays an obvious increase in branching, which is due to the combined effect of excessive AM priming and increased bud activity. Quantitative data show that EXB1 controls the initiation of the AM by positively regulating the transcription of *RAX1*, *RAX2* and *RAX3*. Subsequent data indicate that *EXB1* may be located upstream of the *RAX* gene and regulate AM formation [21,36]. EXB1 also regulates branching in *Arabidopsis* through negative regulation of auxin signaling [37]. Auxin is a well-known bud growth inhibitor, and AM initiation requires minimal auxin [28,38]. In the presence of apical buds exist, auxin is transported from top to bottom in the axilla and inhibits the growth of axillary buds. This phenomenon is called apical dominance [39]. WRKY proteins in the EXB1 clade regulate auxin pathways. Similarly, overexpression of rice *WRKY72* in *Arabidopsis* also increases bud branching, suggesting that the role of EXB1 in bud branching may be evolutionarily conserved between monocots and dicots [40]. Furthermore, a key factor in the establishment of the AM boundary region in *Arabidopsis* is the transcription factor *LATERAL ORGAN BOUNDARYES1* (*LOB1*), which induces *PHYB ACTIVATION TAGGED SUPPRESSOR1* (*BAS1*) expression and encodes a protein that brassinosteroid-inactivation capacity. In leaf axils, BR accumulation is negatively regulated by LOB1, an important boundary-specific transcription factor [18]. LOB1 directly upregulates *BAS1* to produce low BR concentrations to reduce cell division and expansion in border areas [21]. Unlike all other known regulators, AGAMOUS-LIKE 6 (AGL6) specifically promotes stem branching only at the leaf axils of stem leaves in *Arabidopsis* [41].

Similarly, in rice, *TILLERS ABSENT1* (*TAB1*) and *WUSCHEL-RELATED HOMEOBOX4* (*WOX4*) are two *WUS* genes that are necessary for AM initiation [31]. *TAB1* is normally expressed only in the anterior portion of the meristem and is not usually expressed in the SAM or mature AM [42]. Interestingly, *wus* mutation does not affect axillary meristem development in *Arabidopsis*, but rice TAB1 seems to control AM formation through mechanisms different from those of WUS in *Arabidopsis* [43]. The other WUS family transcription factor, WOX4 (the close paralogous homolog of TAB1), plays a role in the development of the AM by alternating with TAB1 [44]. However, unlike *TAB1*, *WOX4* is not expressed in the premeristem but contributes to the maintenance of the AM after almost complete AM establishment. TAB1 forms an AM by enhancing the expression of *O. sativa homeobox 1* (*OSH1*) and *WOX4* [31]. During rice AM formation, *OSH1* is preferentially expressed in the AM, and a significant decrease in its expression and a decrease in tillering in its mutant indicate that the gene is necessary for the initiation or maintenance of the fate of undifferentiated cells at the very early stages of AM formation [45]. *MONOCULM 3* (*MOC3*) is a direct homolog of WUS in rice and is necessary for the formation of tillering buds and interacts with key components of the CK pathway controlling rice tillering [46]. CKs antagonize auxin at the top. Even in the presence of auxin provided by the growing stem apex or the apex, CK applied to buds is sufficient to initiate growth [47].

LAX PANICLE1 (LAX1) and MONOCULM1 (MOC1) encode bHLH family and GRAS family transcription regulators, respectively, which are necessary for the initiation and maintenance of the AM in rice panicles [24,48]. *LAX* is expressed on the boundaries of apical meristem- and neomeristem-forming regions, specifically controlling the initiation or maintenance of new meristems [49]. Studies have shown a significant reduction in the number of spikelets in the *lax1* mutant, and AMs cannot be formed during vegetative development. Similarly, the maize *barren stalk1* (*ba1*) mutant cannot initiate AMs at any stage of the life cycle [24,50]. These two genes accumulate at the proximal axis boundary of the axillary meristems formed during nutrition and reproductive development [27]. *MADS34* in rice encodes a MADS-box transcription factor that coregulates the number of primary and secondary branches with LAX1 [51]. *MOC1* (a direct homolog of *LATERAL SUPPRESSOR*, *LS* in tomato and *LATERAL SUPPRESSOR*, *LAS* in *Arabidopsis*) was the first key transcription factor identified [52,53] controlling rice tillering; it is mainly expressed in leaf axils and axillary buds during AM development, positively regulating rice tillering [43,54]. As expression of *MOC1* and *LAX1* is not altered in the *tab 1* mutant, *TAB1* plays a role in an independent pathway or downstream of MOC1 and LAX1. In addition, LAX2, together with LAX1 and MOC1, plays a role in different AM maintenance pathways to control branching at vegetative and reproductive stages [45]. Similarly, disruption of *LAS* in *Arabidopsis* leads to AM loss during vegetative development, causing loss of branching or tillering and indicating that these genes are highly conserved [5]. In conclusion, both *LAS* and *MOC1* are specifically expressed in the initiation region of the AM [21]. However, in the *moc1* mutant, which lacks axillary buds and tillers, *OSH1* expression completely disappears in leaf axils but is not affected in the SAM. Moreover, *MOC1* is expressed earlier than *LAX1*, *LAX2* and *TAB1* during AM formation, indicating that the sequence or independent role of these genes is the cause of axillary bud formation and that multiple pathways contribute to the development of the AM [31].

Similarly, *REGULATOR OF AXILLARY MERISTEM FORMATION LIKE* (*ROXL*) isolated from sunflower is a homologous gene of *ROX/LAX1*. In situ results of a cross-section show accumulation of *ROXL* transcripts at specific points in the boundary region between the apical meristem and the lateral leaf primordium, displaying a similar pattern in *Arabidopsis* [27]. Based on in situ hybridization, *Ha-ROXL* exhibits clear-boundary transcription in vegetative branches, though the expression pattern of *LATERAL SUPPRESSOR LIKE* (*LSL*) is confined to the boundary region because signals can also be detected in other cellular domains of vegetative and reproductive branches. Transcription of *LSL* is also expanded at the early stage of lateral primordium development, indicating that this gene is involved in the early development of the lateral primordium and in the initiation of the AM [24,55].

*BLIND* (*Bl*) is the homologous gene of the *RAX1* gene in tomato and *Arabidopsis* [21,25]. It encodes an R2R3 Myb transcription factor and regulates the early steps of AM initiation. There are fewer axillary buds due to defects in AM initiation caused by the *bl* mutant [31]. Double-mutation analysis has shown that different members of the Bl-related subgroup of the *R2R3 Myb* gene regulate axillary meristem formation in a partially redundant manner [22]. Double mutants of *ls* and *bl* in tomato display an additive phenotype, suggesting that at least two pathways are involved in AM initiation [20,22,27]. It is noteworthy that these genes are expressed in leaf axils during AM development. The homologous compounds of BLIND and RAX do not play a role in AM formation in rice, indicating that the BLIND/RAX pathway is unique to true dicotyledonous plants [31]. It should also be noted that in rice, axillary meristems usually develop into tillers from the basal nodes of plants, forming a typical cluster plant structure [34]. *FRIZZLE PANICLE* (*FZP*) is a very important gene in rice tillering development. Its overexpression inhibits *RFL/PANICLE ORGANIZATION 2* (*APO2*), which seriously suppresses the formation of axillary meristems at the vegetative stage and leads to a significant decrease in tiller number [48]. In maize, *BAD1* transcripts are detected mainly in the AM boundary region and between lateral branches, and its functional deficiency results in organ fusion via a reduction in the number and angle of branches [56]. Nevertheless, homologs of this gene, *COMPOSITUM 1* (*COM1*) and *RETARDED PALEA1* (*REP1*), in barley and rice have no similar functions. In addition, *BELL1-like homeobox 12* (*BLH12*) and *BLH14* play an important role in maintaining axillary meristems and possess the redundant functions necessary for axillary bud development throughout nutritional and reproductive development [19].

In addition to common plants such as *Arabidopsis*, rice and maize, other species are predicted to have transcription factors involved in branching or tillering. For example, it was found that the formation of the AM in *Antirrhinum majus* requires *ERAMOSA* (*ERA*), a gene encoding a GRAS transcription factor (orthologous to *LAS* in *Arabidopsis*). The basic role of *ERA* in AM formation is consistent with that of *LAS/MOC1/ERA* in preventing cell differentiation in the boundary area and in stimulating AM formation [57].

## 3. Axillary Meristem Outgrowth

The structure of mature plants is determined by the occurrence of axillary meristems, control of bud growth and subsequent dynamics of branching growth. Changes in these parameters result in the high morphological diversity observed in different plant species and even among individuals of a particular species [18]. After the formation of the AM, its growth as a branch repeats the development pattern of primary branches and endows plants with a branching structure [33]. Although a series of genetic studies have revealed the molecular mechanism and genes involved in SAM formation and maintenance, little is known about the generation and growth control of axillary buds [20].

The transcription factor *BRANCHED1* (*BRC1*) in *Arabidopsis* is homologous to *TEOSINTE BRANCHED 1* (*TB1*) in maize and plays a core inhibitory role in regulating axillary bud growth (Figure 3). *BRC2* is a paralog of *BRC1*, which is also expressed in axillary buds and plays a redundant role in the regulation of axillary bud growth [43]. Similarly, *VvBRC* plays a key and negative role in branching in grape [58]. LAS and REV act upstream of *BRC1*. As expression of *BRC1* is significantly downregulated in the *max2* mutant, the MAX-mediated pathway seems to control the activity of BRC1 [59]. *Homeobox 21* (*HB21*), *HB40* and *HB53* act directly downstream of BRC1 to regulate branching [60]. Moreover, BRC1 was identified as an inhibitor of downstream branches of SL signal transduction [61,62]. SL, a plant hormone synthesized by carotenoid catabolism, moves from root to stem, inhibiting stem branching by preventing the growth of axillary buds [63]. In *Arabidopsis*, another transcription factor involved in the SL pathway, namely, BRI1-EMS-SUPPRESSOR1 (BES1), negatively regulates cambium activity in the SL signaling pathway. Further studies have shown that SL signaling regulates the expression level of *WOX4* through BES1, controlling secondary growth [64]. CK is another class of hormones that has an important role in regulating apical dominance and axillary bud outgrowth. At the later stage of development, MYB2 reduces the concentration of CK by inhibiting the expression of *IPTs*, suppressing the growth of axillary buds [47]. In functional exploration of the MADS domain factor *FRUITFULL* (*FUL*) in *Arabidopsis thaliana*, it was found that the combination of auxin and BR strongly induced the growth regulator *SMALL AUXIN UPREGULATED RNA 10* (*SAUR10*) to be directly modulated by FUL, thus participating in branching angle regulation [65]. It should be noted that FUL also responds to a decrease in R:FR light by regulating the SAUR10 pathway and affecting *Arabidopsis* branching [65]. The auxin-regulated branching pathway in *Arabidopsis* involves a class of photochrome-targeted transcription factors, *PHYTOCHROME INTERACTING FACTOR 4* (*PIF4*)/*PIF5* [66]. When these transcription factors participate in photoreactions, they inhibit the branching caused by phyB dysfunction and low R: FR. Hence, R:FR plays an important role in branching.

Overall, the interaction between ABA and auxin may mediate the effect of PIF4/PIF5 on branching [67]. Mohammad et al. [68] reported another transcription factor, ERF BUD ENHANCER (EBE), that affects cell proliferation, axillary bud growth and branching in *Arabidopsis*. The gene encodes the AP2/ERF transcription factor and is strongly expressed in proliferating cells. Moreover, overexpression of *EBE* promotes cell proliferation, shortens the cell cycle in growing calli and stimulates the formation and growth of axillary buds [68].

*TB1*, also known as *FINECULM1* (*FC1*), is a TCP family transcription factor that negatively regulates rice tillering and inhibits the subsequent growth of axillary buds [69]. This gene encodes a TCP TF that is expressed at the base of these buds and SAMs. Its overexpression leads to a significant reduction in tiller number, whereas the *tb1* mutant exhibits an increase in tiller number [31]. In addition, TB1 may be a common target for the CK and SL pathways and act downstream of SL [70,71]. *MADS57* is a transcription factor of the MADS domain family that participates in the regulation of axillary bud growth in rice. RNA in situ hybridization analysis has shown that MADS57 is mainly expressed in meristems and axillary buds. Additionally, its expression is higher in the tillering and stem elongation stages than in other stages of rice growth [70]. Interaction of TB1 with MADS57 reduces the negative regulatory activity of MADS57 on *D14* expression, allowing MADS57 to affect tillering [71].

In a study of branching-related genes in chrysanthemum, *CmERF053* was found to be rapidly upregulated in axillary buds when apical dominance was relieved, which may be related to the growth of lateral branches. The gene belongs to the AP2/ERF family and is mainly expressed in stem and root organs. Further transcriptome analysis showed that CK-mediated control of stem branching may be related to the transcription level of *CmERF053* [26]. *IDEAL PLANT ARCHITECTUREL1* (*IPA1*), also known as *SQUAMOSA PROMOTER BINDING PROTEIN-LIKE 14* (*SPL14*), is another key regulator that determines plant structure but not meristem activity, and upregulation of *IPA1* expression leads to fewer tillers in rice [72]. Furthermore, *IPA1* overexpression lines display a reduced tillering phenotype, whereas tillering in the *ipa1* mutant increases [73]. This gene acts as a direct downstream component of D53 in regulating tiller number and SL-induced gene expression. In fact, *D53* is one of the only known transcription targets of SL, and D53 inhibits *IPA1* upregulation [74]. IPA1 directly binds to the *D53* promoter and plays a key role in the feedback regulation of SL-induced *D53* expression. In summary, *IPA1* may constitute a long-term transcription factor that acts with D53 to mediate SL-regulated rice tillering development [75]. In later vegetative axillary bud growth, OsMADS57 enhances axillary bud growth and subsequent tillering through SL signal transduction via direct inhibition of expression. OsMADS57 directly inhibits *D14* transcription to regulate tillering during organogenesis [70].

The SQUAMOSA PROMOTER BINDING PROTEIN-LIKE 13 (SPL13) gene encodes an SBP transcription factor that is mainly expressed in meristems and is essential for regulating branching and vegetative growth in *Medicago* [63]. Overexpression of *SPL13* inhibits axillary bud growth, thereby reducing lateral branches. In this process, *MYB112* is targeted and downregulated by SPL13. Compared with WT, *MsMYB112* RNAi plants show more branches, which confirms that MYB112 itself also inhibits the lateral branch growth of alfalfa [63,76]. Furthermore, MdWUS2 in apple can regulate branches by inhibiting the expression of *MdTCP12* [77].

*Related to ABI3 and Viviparous 1* (*RAV1*) is a circadian rhythm gene in *Castanea mollissima* that is homologous to the *TEM* gene in *Arabidopsis*. It reaches its peak expression at noon under vegetative growth conditions and is highly expressed during winter dormancy and in response to low temperature [78]. When the CsRAV1 protein was overexpressed in hybrid poplar, high sylleptic branching was induced during the same growing season as lateral bud formation.

## 4. Application of Branching-Regulating TFs

Genetic control of branches is the main determinant of yield, seed number regulation and harvestability. For example, interference mutations in *FZP* and *IPA1* in rice, *TB1* in maize, and *BRC* in grape lead to increased branching and increasing yield [20,48,75,79]. Overexpression of *Bl* in tomato increases the number of branches, also increasing yield. At the same time, the change in plant type caused by branching provides an opportunity to explore the ornamental value of plants. For example, overexpression of *CmERF053* significantly increases the number of branches in chrysanthemum. In summary, regulation of these key transcription factors can significantly increase the number of branches, thereby increasing crop yield or quality.

In addition, fine regulation of branching has become an important strategy for plants to morphologically adapt to various environments. Chestnut CsRAV1, a circadian rhythm response factor, participates in the winter dormancy and low-temperature response of poplar and increases branching. Thus, manipulating this gene may lead to the possibility of producing trees with greater biomass. In actual cultivation, plants usually inhibit axillary bud growth in response to reductions in the ratio of red light to far-red light (R:FR) caused by the presence of competitive neighbors. Overexpression of *PIF4* and *PIF5* significantly inhibit the branching caused by this shade-avoidance syndrome, providing opportunities for practical cultivation.

## 5. Environmental Pathways Involved in the Control of Shoot Branching

It is well known that plant types have remarkable plasticity. Branch development is affected by many external factors, such as light, temperature and soil nutrients. Light is a powerful environmental factor that affects the branching of plants [80]. For example, low-intensity light reduces tillering in *Triticum aestivum* [81], whereas high-intensity light increases branching in hybrid roses [82]. Low R:FR and a decrease in blue light intensity trigger SAS, which leads to a decrease in axillary bud growth ability, such as in *Rhododendron* and *Hordeum vulgare* [83]. However, in *Lilium*, FR light strongly inhibits bud outgrowth [84], and blue light can increase or decrease the length of branches and internodes [85]. In general, UV radiation exposure reduces the length of branches [86], and studies have shown that photoperiod is one of the environmental factors involved in regulating branching, altering the branching pattern [87]. In summary, light is of great significance for the regulation of branching. In addition to light, temperature, moisture, carbon dioxide and other environmental factors affect the branching of plants. High temperature can inhibit branching, and CO_2_ reduces the negative impact of high temperature on branch growth [88]. Of course, water and nutrients (such as nitrogen and phosphorus) are decisive factors in regulating plant shoot branching [89]. Many TFs responding to plant stress responses have been reported. However, research on TFs involved in branch response stress is scarce. Therefore, further exploration of TFs involved in branching response stress is worthy of attention.

## 6. Perspectives

Branching determines plant architecture and crop yield and plays an important role in plant morphogenesis. Therefore, research on branching regulation mechanisms is a popular topic worldwide. The regulation of plant branching by different transcription factors through mutual connections is one of the main directions of the current study of branching development patterns. Previous research and discussion on a single transcription factor have been the premise and foundation for studying the transcription regulatory network.

Notably, the functions of some transcription factors are not conserved. For example, *RAV1* in chestnut is a circadian rhythm gene that is homologous to the *TEM* gene in *Arabidopsis*. However, the two genes may lead to different phenotypes in woody and herbaceous plants. In view of the transcription factors related to branching discovered to date, we found exploration of the new functions of known transcription factors to be innovative, even though conserved transcription factors appear to provide key targets for the branching regulation mechanism.

## Figures and Tables

**Figure 1 plants-11-01997-f001:**
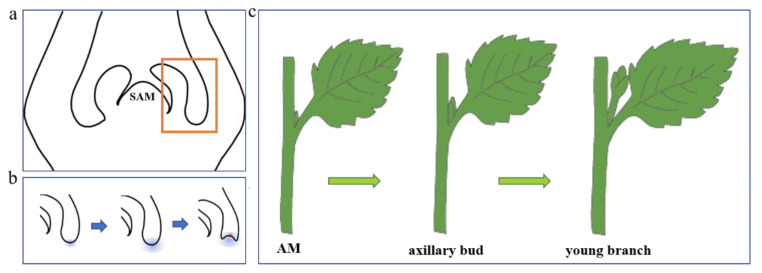
Steps of plant shoot branching. (**a**) indicates the axillary meristem at the leaf primordium axils. (**b**) indicates the formation of axillary meristems (area shown in yellow box in (**a**)). (**c**) represents the development of plant axillary meristem into young branches.

**Figure 2 plants-11-01997-f002:**
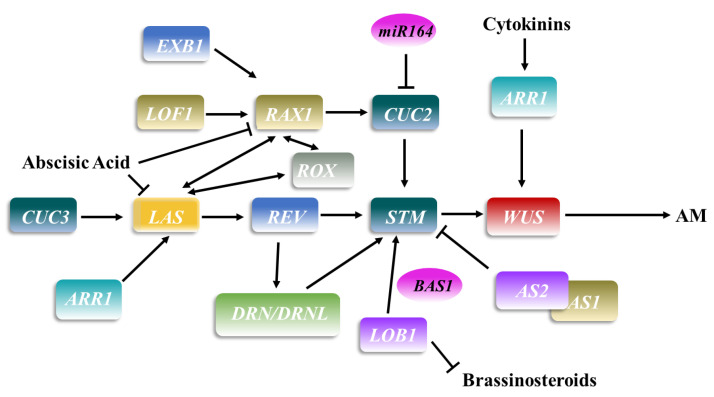
The pattern of transcription factors involved in regulating axillary meristem formation in *Arabidopsis thaliana*. The rectangular box represents a transcription factor, the same color represents the same family, and ellipses represent genes other than transcription factors. The arrow and rough line represent positive and negative regulation, respectively.

**Figure 3 plants-11-01997-f003:**
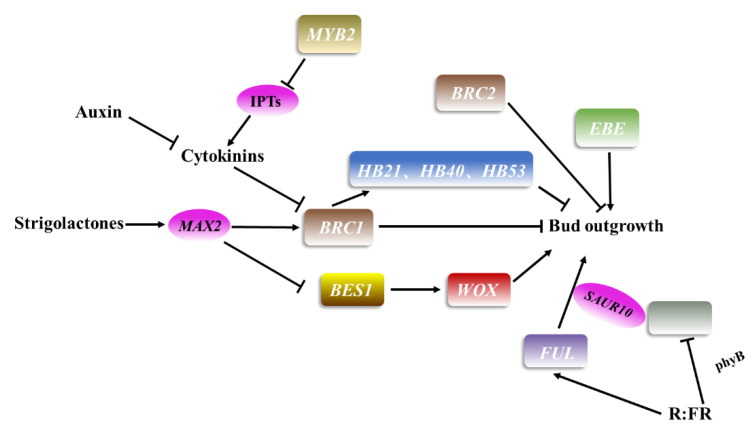
Patterns of transcription factors involved in regulating axillary meristem growth in *Arabidopsis thaliana*. The rectangular box represents transcription factors, the same color represents the same family, and ellipses represent genes other than transcription factors. The arrow and rough line represent positive and negative regulation, respectively.

**Table 1 plants-11-01997-t001:** Transcription factors involved in regulating plant branching.

Name	Homologs in Other Species	Family	Function
AtAGL6 (AGAMOUS-LIKE6)		MADS	Facilitates the formation of axillary meristems
OsMADS34			Coordinates with LAX1 to regulate the number of primary branches
OsMADS57			Is expressed predominantly in the SAM and axillary buds and is involved in SL signaling to enhance axillary bud growth and subsequent tillering
AtFUL (FRUITFULL)			Is involved in development of the axillary meristem, the expression of which is controlled by auxin
AtCUC1-3 (CUP SHAPED COTYLEDON1-3)		NAC	Is negatively regulated by BRs and involved in AM initiation
AtSTM (SHOOT MERISTEMLESS)	OsOSH1 (*O. sativa* homeobox1)	HB-KNOX	Is involved in initiation or maintenance of undifferentiated cell fate in very early stages of AM formation
AtLOF1 (LATERAL ORGAN FUSION1)		MYB	Is involved in lateral organ separation and axillary meristem formation
AtLOF2 (LATERAL ORGAN FUSION2)			
AtAS1 (ASYMMETRIC LEAVES1)			Inhibits branching and downregulates STM when cells start to differentiate
AtRAX1 (REGULATOR OF AXILLARY MERISTEMS 1)	Bl (Blind), *S. lycopersicum*		Is involved in the early steps of AM initiation and development
AtRAX2-3			
MsMYB112			Inhibits collateral growth
AtMYB2			Inhibits branching and reduces cytokinin concentrations by inhibiting expression of IPTs in *Arabidopsis*
AtWUS (WUSCHEL)	OsTAB1 (TILLERS ABSENT1); OsMOC3 (MONOCULM 3)	WOX	Promotes branching and is involved in maintenance of meristematic stem cell function and regulation of cell division
OsWOX4			Is involved in AM initiation
AtWOX4			Regulates *Arabidopsis* secondary growth by SL signaling
MdWUS2 (WUSCHEL 2)			Regulates branching by inhibiting the activity of MdTCP12 (BRC2 homolog)
AtLAS (LATERAL SUPPRESSOR)	Ls, *S. lycopersicum*; ERA (ERAMOSA, *A. majus*); OsMOC1; HaLSL (LATERAL SUPPRESSOR LIKE)	GRAS	Is necessary for maintenance of the meristematic potential of the cells in the axils of leaf primordia
HaROXL (REGULATOR OF AXILLARY MERISTEM FORMATION LIKE)	ZmBA1 (BARREN STALK1); OsLAX1 (Lax Panicle 1); AtROX (REGULATOR OF AXILLARY MERISTEM FORMATION)	bHLH	Is involved in development of the SAM and lateral young leaf primordia
OsLAX2 (LAX PANICLE2)			Is involved in development of the SAM and lateral young leaf primordia
AtPIF4/5 (PHYTOCHROME INTERACTING FACTORs 4/5)			Inhibits the branching caused by phyB dysfunction and low R:FR
OsFZP (FRIZZLE PANICLE)	ZmBD1; COM2, *H. vulgare*	AP2/ERF-ERF	Represses axillary meristem formation
AtEBE (ERF BUD ENHANCER)			Is involved in cell proliferation and axillary bud growth
AtDRN (DORNRÖSCHEN)			Regulates *STM* expression and AM initiation
AtDRNL (DORNRÖSCHEN-LIKE)			
AtERF053			Is involved in cytokinin control of stem branching
OsRFL (RICE FLORICULA/LEAFY)	OsAPO2 (PANICLE ORGANIZATION 2)		Promotes AM specificity through its action on LAX1 and CUC genes
ZmBAD1 (BRANCH ANGLE DEFECTIVE1)		TCP	Promotes the formation of lateral meristems (e.g., branches) and axillary organs (e.g., leaf pillows) in wild-type maize
AtBRC1 (BRANCHED1)	OsTB1 (TEOSINTE BRANCHED 1); OsFC1 (FINECULM1); VvBRC		Negatively regulates axillary bud growth
AtBRC2 (BRANCHED2)	MdTCP12		Has a redundant role with BRC1 in regulation of axillary bud growth
AtAS2 (ASYMMETRIC LEAVES1)		LOB	Inhibits branching and downregulates STM when cells start to differentiate
AtLOB1 (LATERAL ORGAN BOUNDARIES 1)			Is negatively regulated by BRs to reduce cell division and expansion in the border zone
AtWRKY71/EXB1		WRKY	Is expressed in tissues surrounding the AM start site
WRKY72			Positively regulates bud branching
AtREV (REVOLUTA)		HD-ZIP	Upregulates *STM* expression and promotes AM initiation
HB21 (Homeobox21)			Inhibits branching, directly downstream of BRC1
HB40 (Homeobox40)			
HB53 (Homeobox53)			
AtSPL13 (SQUAMOSA PROMOTER BINDING PROTEIN-LIKE 13)		SBP	Inhibits the growth of axillary buds
AtIPA1 (IDEAL PLANT ARCHITECTUREL1)	OsSPL14		Acts with D53 to mediate SL-regulated tiller development in rice
AtARR1 (ARABIDOPSIS RESPONSE REGULATOR 1)		GARP-ARR-B	Promotes branching, acts Downstream of cytokinins and promotes LAS expression by binding to their promoters
AtBES1 (BRI1-EMS-SUPPRESSOR1)		BES1	Inhibits branching and negatively regulates cambium activity in the SL signaling pathway in *Arabidopsis*

Note: The transcription factor prefix indicates the species to which it belongs. At—Arabidopsis thaliana; Os—Oryza sativa; Ha—Helianthus annuus; Cr—Ceratopteris richardii; Zm—Zea mays; Sl—Solanum lycopersicum; Md—Malus pumila; Vv—Vitis vinifera.

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
