# Peer review of "The Role of Transcription Factors in the Regulation of Plant Shoot Branching"

_plants, 2022, doi:10.3390/plants11151997_

Round 1

Reviewer 1 Report

The improved review is well written and informative. It provides an extensive overview about plant transcription factors that determine branching. I only found that reference [83] is missing from the list.

Author Response

1.The improved review is well written and informative. It provides an extensive overview about plant transcription factors that determine branching. I only found that reference [83] is missing from the list.

A: Thank you for your useful advice. We have added the reference in line 612-613.

Reviewer 2 Report

Dear Authors,

the Introduction content and intent is good, but the English is especially poor. This section requires a complete review and edit by someone who has better English and grammar skills.

E.g: line 10 "....balance plant ..."  I beg you to rephrase that "...play a key role during..."

Transcription factors, also Known as "... trans-acting farcors" or "as sequence-specific DNA binding factors..."?

Line 36 please, change "Lateral branch development..." with  "shoot"

Line 36- 37 - you need to reference this section (Shoot branching is an important feature of plant architecture, for growth of ornamental plants; Branching also influences the plant competitiveness against weeds or the propagation of pests ...)

Author Response

1.the Introduction content and intent is good, but the English is especially poor. This section requires a complete review and edit by someone who has better English and grammar skills.

A: Thank you for your useful advice. We have polished the language in the article.

2.E.g: line 10 "....balance plant ..."  I beg you to rephrase that "...play a key role during..."

A: Thank you for your useful advice. We have changed the sentence in line 9-10.

3.Transcription factors, also Known as "... trans-acting farcors" or "as sequence-specific DNA binding factors..."?

A: Thank you for your useful advice. We have changed the sentence in line 25.

Line 36 please, change "Lateral branch development..." with "shoot"

A: Thank you for your useful advice. We have changed the sentence in line 35.

4.Line 36- 37 - you need to reference this section (Shoot branching is an important feature of plant architecture, for growth of ornamental plants; Branching also influences the plant competitiveness against weeds or the propagation of pests ...)

A: Thank you for your useful advice. We have added the reference in line 36-37.

References

  1. Acker, R.; Weise, S. F.; Swanton, C. J. Influence of interference from a mixed weed species stand on soybean (Glycine max (l.) merr.) growth. Canadian Journal of Plant Science 1993, 73(4), 1293-1304.
  2. Costes, E.; Lauri, P. E.; Simon, S.; Andrieu, B. Plant architecture, its diversity and manipulation in agronomic conditions, in relation with pest and pathogen attacks. European Journal of Plant Pathology 2013, 135, 455-470.
